# Distribution of Monocarboxylate Transporters in Brain and Choroid Plexus Epithelium

**DOI:** 10.3390/pharmaceutics15082062

**Published:** 2023-07-31

**Authors:** Masaki Ueno, Yoichi Chiba, Ryuta Murakami, Yumi Miyai, Koichi Matsumoto, Keiji Wakamatsu, Genta Takebayashi, Naoya Uemura, Ken Yanase

**Affiliations:** 1Department of Pathology and Host Defense, Faculty of Medicine, Kagawa University, Takamatsu 761-0793, Kagawa, Japan; chiba.yoichi@kagawa-u.ac.jp (Y.C.); murakami.ryuta@kagawa-u.ac.jp (R.M.); miyai.yumi@kagawa-u.ac.jp (Y.M.); matsumoto.koichi@kagawa-u.ac.jp (K.M.); s20d727@kagawa-u.ac.jp (K.W.); 2Department of Anesthesiology, Faculty of Medicine, Kagawa University, Takamatsu 761-0793, Kagawa, Japan; takebayashi.genta@kagawa-u.ac.jp (G.T.); uemura.naoya@kagawa-u.ac.jp (N.U.); yanase.ken@kagawa-u.ac.jp (K.Y.)

**Keywords:** choroid plexus, epithelial cell, lactate, monocarboxylate transporter, thyroid hormone

## Abstract

The choroid plexus (CP) plays central roles in regulating the microenvironment of the central nervous system by secreting the majority of cerebrospinal fluid (CSF) and controlling its composition. A monolayer of epithelial cells of CP plays a significant role in forming the blood–CSF barrier to restrict the movement of substances between the blood and ventricles. CP epithelial cells are equipped with transporters for glucose and lactate that are used as energy sources. There are many review papers on glucose transporters in CP epithelial cells. On the other hand, distribution of monocarboxylate transporters (MCTs) in CP epithelial cells has received less attention compared with glucose transporters. Some MCTs are known to transport lactate, pyruvate, and ketone bodies, whereas others transport thyroid hormones. Since CP epithelial cells have significant carrier functions as well as the barrier function, a decline in the expression and function of these transporters leads to a poor supply of thyroid hormones as well as lactate and can contribute to the process of age-associated brain impairment and pathophysiology of neurodegenerative diseases. In this review paper, recent findings regarding the distribution and significance of MCTs in the brain, especially in CP epithelial cells, are summarized.

## 1. Introduction

### 1.1. Barrier Function in Choroid Plexus

The brain restricts the entrance of ions and solutes circulating in the blood by two cellular barriers: the blood–brain barrier (BBB) and blood–cerebrospinal fluid (CSF) barrier (BCSFB) [1,2,3,4]. The BBB is composed of endothelial cells interconnected by tight junctions, two basement membranes, pericytes, and end-feet of astrocytes [1,5,6,7,8,9]. BBB endothelial cells have few vesicles and no fenestrations in the cytoplasm, whereas the endothelial cells are equipped with various transporters to supply energy from the blood to brain cells, such as (a) carbohydrate transporters, (b) monocarboxylate transporters (MCTs), (c) amino acid transporters, (d) fatty acid transporters, (e) nucleotide transporters, (f) hormone transporters, (g) organic anion and cation transporters, and (h) other transporters, including those for amines and choline [2,3,4,9]. However, barrier and carrier functions of BBB are affected by various invasions in disease states [7]. Accordingly, BBB dysfunction leads to various brain dysfunctions, such as cognitive dysfunction. On the other hand, endothelial cells of capillaries in the choroid plexus (CP) are originally fenestrated, allowing the passage of intravascular low-molecular-weight substances as well as ions in CP with a vascularized stroma [8,10,11,12]. Every CP has a vascularized stroma and is covered by a monolayer of epithelial cells interconnected by tight and adherens junctions. The tight junction contains occludins and claudins, which bind to cytosolic zonula occludens protein-1 (ZO-1) [10,11]. On the other hand, the adherens junction is distributed beneath the tight junction and contains cadherins, which bind to catenins distributed along the lateral surface of the cytoplasm of CP epithelial cells (CPEs) [10,11]. Accordingly, neighboring CPEs serve as a barrier between the blood and CSF, referred to as BCSFB, and restrict the entry of intravascular substances into ventricles [8,9].

### 1.2. Characterization of Choroid Plexus Epithelial Cells

CSF is produced through several kinds of ion and water transporters located in CPEs [1,10,11,13]. CPEs are characterized by the presence of various epithelial cytokeratins, β-catenin, vimentin, S-100 protein, podoplanin, and transthyretin/prealbumin [10,11,13,14,15,16]. Intermediate filaments in CPEs have been identified as keratins 8, 18, and possibly 19 [10,17,18]. Miettinen et al. [18] revealed immunoreactivity for cytokeratin (CK) 19 in CP tissues of human brains by immunoblotting, whereas Kasper et al. [17] found no immunoreactivity for CK 19 in CPEs of human brains by immunohistochemistry. We could not confirm immunoreactivity for CK19 (Progen, Biotechnik GmbH, Heidelberg, Germany, 61010) in human CPEs, although immunoreactivities for CK8 and CK18 were confirmed to be present in CPEs (Figure 1a–c). Justice et al. reported that strong immunoreactivity for α-1-antichymotrypsin was present in apical granular organelles in the cytoplasm of adult CPEs [19]. We confirmed, as shown here by immunostaining, that β-catenin, vimentin, S-100, podoplanin, transthyretin, and α-1-antichymotrypsin are expressed in the cytoplasmic membrane or cytoplasm of human CPEs (Figure 1d–i). In addition, representative transporters related to CSF production, such as Na^+^, -K^+^, -ATPase, aquaporin 1 (AQP1), and anion exchange protein 2 (AE2), are present in apical or basolateral cytoplasmic membranes of CPEs (Figure 1j–l) [10,11,12,13].

BBB and BCSFB not only have barrier functions but also carrier functions for intracerebral transport of essential nutrients such as glucose, lactate, and amino acids [1,2,3,4,9]. In addition, they play important roles in the removal of metabolic wastes and neurotoxic substances such as amyloid β (Aβ) [1,2,8,20,21]. It is well-known that one of the significant functions of CP is to produce and secrete CSF [11,20]. Accordingly, CPEs are equipped with transporters for ions and organic solutes that are different from those in BBB endothelial cells [1,2,8,21,22]. We previously reviewed the distribution of transporters for glucose, fructose, and urate [23]. Accordingly, in this review paper, we focused on recent findings on the distribution of MCTs in the brain, especially in CPEs. In the brain, MCTs 1, 2, 3, and 4 transport lactate, pyruvate, and ketone bodies, whereas MCT8 transports thyroid hormones. In this manuscript, previously reported results on the distribution of these MCTs in CPEs were summarized and confirmed by immunohistochemical staining. Then, their physiological function and the pathophysiological importance of their expression at BCSFB in neurological diseases were discussed. Table 1 shows a summary of clinicopathological profiles of autopsied human brains that were removed at Kagawa University Hospital, as introduced in our previously published papers [24,25]. Table 2 shows a summary of antibodies used in this manuscript. Before incubation with some antibodies, antigen retrieval was performed by heating sections in 10 mM sodium citrate buffer (pH 6) or 1 mM tris(hydroxymethyl)aminomethane (Tris)-ethylenediaminetetraacetic acid (EDTA) buffer (pH 9) for 20 min. These studies using autopsied human brains were approved by the institutional Ethics Committee of the Faculty of Medicine, Kagawa University, in accordance with the Declaration of Helsinki [24,25]. Then, these brain samples were used to confirm the distribution of substances, including MCTs, in this review manuscript.

## 2. Monocarboxylate Transporters in the Brain

MCTs catalyze the proton-linked influx or efflux of monocarboxylates such as L-lactate, pyruvate, and ketone bodies in various cells of several organs [26,27]. Consequently, MCTs enable the 1:1 exchange of monocarboxylate and protons across the cellular membrane [26,27]. The direction of transport depends on their intracellular and extracellular concentrations [27]. It is well-known that there are four isoforms, MCTs 1, 2, 3, and 4, in the brain. They belong to the SLC16 family of solute carriers, which has 14 members in total. The family includes MCTs to transport thyroid hormones. MCT8, which is a high-affinity transporter for 3,5,3′-triiodothyronine (T3), MCT10, which is the most homologous to MCT8, and eight orphan members also constitute the family [28]. MCTs 1–4 catalyze proton-coupled lactate transport, whereas MCT8 and MCT10 catalyze the sodium- and proton-independent transport of thyroid hormones [28,29]. In the brain, it is known that MCT1, MCT2, and MCT4 are widely expressed in several kinds of cells [30,31]. MCT1 is mainly expressed on endothelial cells with the barrier function both in humans and rodents [32,33]. MCT2 and MCT4 are mainly expressed in neurons and astrocytes, respectively. On the other hand, MCT3 is expressed in the retinal pigment epithelium and CPEs of mice [34,35]. Protein expression levels in plasma membrane fractions of isolated CP of humans and rats were measured using quantitative targeted absolute proteomics [36]. The study identified low-level expression of MCT1 and MCT3 in rats, low-level expression of MCT1 in humans, and very-low-level expression of MCT4 and MCT5 in humans [36]. It is likely that the expression level of transporters for lactate, pyruvate, and ketone bodies in CPEs affects their CSF concentrations. Their values in CSF have been reported in various diseased brains. Glucose transporters belong to either SLC2A/GLUT or SLC5A/SGLT families and are summarized in some papers [23,37,38]. In this review manuscript, the distributions of MCT1, MCT2, MCT3, MCT4, MCT5, and MCT8, which were previously reported to be present in CPEs, were first reviewed. In addition, detailed localization of MCTs in the brain, especially in cerebral microvessels and CPEs, was confirmed by immunohistochemical staining, as the microvessels and CPEs are important routes to transport intravascular substances from the blood into the brain.

### 2.1. MCT1 (SLC16A1) Distribution in the Brain and CPEs

MCT1 is well-known to be expressed ubiquitously in the brain. MCT1 expression was reported in endothelial cells, astrocytes [39,40,41], and oligodendrocytes [42]. Lactate is released from astrocytes via MCT4 and may be carried into oligodendrocytes via MCT1 [42]. It was reported that MCT1 was immunohistochemically expressed in microglia in healthy human brains [40] and on the apical side of the cytoplasm of CPEs in autopsied diseased human brains [43]. Immunoreactivity for MCT1 (Abcam, ab90582) is confirmed to be expressed in endothelial cells, reactive astrocytes, and on the apical side of the cytoplasm of CPEs (Figure 2a–c).

### 2.2. MCT2 (SLC16A7) Distribution in the Brain and CPEs

MCT2 catalyzes the proton-coupled transport of many monocarboxylates, including lactate, pyruvate, and ketone bodies, across the plasma membrane. MCT2 shows the highest affinity for lactate [29] and is also a high-affinity pyruvate transporter. The MCT2 gene is known to be transcribed with high-sensitivity in response to hypoxia, intracellular pH, and lactate [44]. MCT2 is expressed mainly in neurons and also in astrocytes [45]. Pierre et al. [46], using immunohistochemical techniques, reported that MCT1 was strongly expressed in astrocytes of mice, whereas MCT2 was expressed in a small subset of neurons of mice. These findings are consistent with the concept that lactate is released by astrocytes via MCT1 and is taken up into neurons via MCT2. They also reported [46] that CPEs of mice were heavily immunostained for MCT2 as well as MCT1. On MCT2 expression in human brains, immunoreactivity for MCT2 was present in neuronal axons, microglia, and endothelial cells in healthy human brains and additionally in astrocytes in brains of multiple sclerosis patients [41]. Immunoreactivity for MCT2 (Abcam, ab198272) is confirmed to be present in neuronal cytoplasm, endothelial cells, and reactive astrocytes, and on the apical side of the cytoplasm of human CPEs, as shown in Figure 2d–f.

### 2.3. MCT4 (SLC16A3) Distribution in the Brain and CPEs

MCT4 is a low-affinity high-capacity transporter and is expressed mainly in astrocytes [47]. MCT4 is known to facilitate the excretion of lactate in cells, in which glycolysis is highly active [48]. Immunoreactivity for MCT4 was present in microglia and endothelial cells as well as astrocytes of healthy human brains [41]. Murakami et al. [43] reported that MCT4 immunoreactivity was present in endothelial cells and reactive astrocytes, and on the basolateral side of the cytoplasm of CPEs in diseased human brains. Immunoreactivity for MCT4 (Abcam, ab244385) can be noted in the same location as reported previously (Figure 2g–i).

### 2.4. MCT3 (SLC16A8) Distribution in CPEs

MCT3 protein and mRNA of mice were detected in the retinal pigment epithelium and CPEs by immunohistochemistry and Western and Northern blot analyses [34]. Immunoreactivity for MCT3 was noted in the basolateral membrane of CPEs of mice. MCT3 expression was also reported to be detected in rat CP but not in human samples by quantitative targeted absolute proteomics [36]. MCT3 immunoreactivity (Abcam, ab60333) is seen in the cytoplasm of CPEs but not in microvessels (Figure 2j,k).

### 2.5. MCT5 (SLC16A4) Distribution in the Brain

The expression of MCT5 in the brain was reported in a paper by Halestrap et al. [26] and also shown in the isolated human CP by quantitative targeted absolute proteomics [36]. Beckner et al. [49] reported that MCT5 expression is increased in some glioblastoma cells. Immunoreactivity for MCT5 (Abcam, ab191008) is not seen in the diseased human brain, including CPEs (Figure 2l,m).

### 2.6. MCT8 (SLC16A2) Distribution in the Brain and CPEs

MCT8, also known as SLC16A2, was first cloned in 1994 and called XPCT because it was encoded by the XPCT gene in Xq13.2. [50]. After MCT10 was identified to be originally an aromatic amino acid transporter, MCT8 was established as an active transporter to transport iodothyronines, including the thyroid hormones T3 and T4 [51]. It is now considered that MCT8, MCT10, and organic anion transporting polypeptide 1C1 (OATP1C1) are the best-characterized specific thyroid hormone transporters [28,52]. MCT8 is widely expressed in most tissues, including the liver, kidney, heart, skeletal muscle, brain, pituitary, and thyroid [53]. It plays a major role in thyroid hormone uptake across the BBB [54]. In brains of humans and mice, MCT8 protein was reported to be expressed in the cortex, hippocampus, cerebellum, hypothalamus, tanycytes, cerebral vessels, and CP [55,56,57]. Roberts et al. [55] reported that MCT8 was immunohistochemically expressed in endothelial cells and was also visible on the apical and basal surfaces of human CPEs, whereas immunoreactivity for OATP-14, known as OATP1C1, was present on both apical and basolateral surfaces of CPEs. On the other hand, Roberts et al. stated [55] that MCT8 is expressed on the apical surface of CPEs and OATP14 is present primarily on the basolateral surface of CPEs in human and rodent brains. Wilpert et al. [58] reported that MCT8 protein in human brains was expressed in endothelial cells of BBB, CPEs, and tanycytes, whereas neuronal MCT8 protein was expressed in large quantities in specific brain regions. Alkemade et al. [59] reported that MCT8 was immunohistochemically expressed in neurons and glial cells of the human hypothalamus. In contrast, in mouse brains, MCT8 mRNA was expressed predominantly in neurons and also in CP [60]. Alkemade et al. [61] reported that MCT10 immunocytochemical staining was noted in neurons of hypothalamic nuclei. Mutations of MCT8 cause a severe neurodevelopmental disorder, Allan–Herndon–Dudley syndrome [62], which is an X-linked inherited disorder of brain development with hypomyelinating leukodystrophy. Patients developed several kinds of symptoms, such as hypotonia, primitive reflexes, scoliosis, muscular hypoplasia, and dystonia [62,63]. These indicate that thyroid hormone and MCT8 are essential for nervous system development. Immunoreactivity for MCT8 (Novus, NBP2-57308) is present in endothelial cells and on the apical side of the cytoplasm of CPEs (Figure 2n,o). Confirmatory immunohistochemical images in this manuscript are shown in Figure 1 and Figure 2, whereas previously reported findings on regional distribution and cellular localization of MCTs in many papers are summarized in Table 3.

### 2.7. Lactate Transport in the Brain through Cerebral Microvessels and CPEs

MCT1 and MCT4 are involved in lactate release by astrocytes and contribute to energy supply by glycolysis in the cells [64,65,66]. MCT4 is a transporter with low affinity and high capacity for lactate and contributes to transport lactate from astrocytes to neurons. In contrast, MCT2 is mainly present in neurons, and lactate is taken up into neurons via MCT2 as an efficient oxidative energy substrate. MCT2 is known to be a transporter with a higher affinity for most monocarboxylates than MCT1 [26]. These MCTs contribute in concert to the astrocyte–neuron lactate shuttling [67,68,69]. Accordingly, the distribution of MCTs in the plasma membrane of neurons and astrocytes suggests a significant role of these transporters in the shuttling of energy metabolites between neurons and astrocytes [67,68,69,70].

It was reported that lactate values in CSF of 7614 individuals increased with aging [71]. Results from a CSF-based study indicated that lactate levels in CSF of Parkinson’s disease (PD) patients increased compared with controls and were correlated with clinical disease progression [72]. On the other hand, lower lactate levels in CSF were reported in patients with dementia, including Alzheimer’s disease (AD) and frontotemporal dementia, compared with non-demented individuals [73]. Accordingly, lactate levels in CSF may be useful for understanding the degree of aging and progression of neurodegenerative diseases. The hydroxy-carboxylic acid 1 receptor (HCA1 receptor), a receptor for lactate, is highly expressed in principal neurons, whereas the receptor is also expressed in astrocytes and endothelial cells [35,74]. In addition, it was recently reported that the HCA1 receptor was expressed in CPEs [43]. Immunoreactivity for the HCA1 receptor (Novus, NLS-2095) is shown to be present in cerebral endothelial cells and on the basolateral membrane of CPEs (Figure 2p,q).

## 3. Discussion

In this manuscript, we first reviewed the localization and significance of MCTs in cerebral microvessels and CP, and subsequently confirmed the detailed localization of cytoplasmic and membranous molecules reported previously to be expressed in CPEs. As lactate is transported with protons through MCTs in CPEs, representative transporters for water and electrolytes are also described in this review paper. Although MCT5 was not immunohistochemically confirmed to be localized in CPEs, MCT1, MCT2, and MCT4, which are the main transporters for lactate in the brain, were shown to be immunohistochemically expressed in CPEs as well as astrocytes and endothelial cells. MCT2 was also expressed in the cytoplasm of some neurons. MCT3 was shown to be immunohistochemically expressed in the cytoplasm of CPEs. In addition, the HCA1 receptor, which is a receptor for lactate and mediates a decrease in cellular cAMP levels, was immunohistochemically expressed in cerebral microvessels and CPEs. Polarized distributions of MCT1, MCT2, MCT3, MCT4, and HCA1-R and putative directions of lactate through CPEs are indicated by dashed arrows in Figure 3. Considering the movement of lactate between astrocytes and neurons via MCTs, it can be suggested that lactate is transported from CSF into the cytoplasm of CPEs via MCT2 and is released to the CP stroma via MCT4, whereas MCT1 may facilitate the transport of lactate from the cytoplasm of CPEs into CSF (Figure 3). At present, however, the directions cannot be determined. It remains to be clarified whether MCT3 is involved in the transport of lactate in CPEs.

Lactate values in CSF are known to increase with aging [71]. Lactate levels in PD patients increased compared with those of controls [72], whereas lower CSF lactate levels were reported in patients suffering from dementia, including AD, compared with non-demented individuals [73]. It is likely that excess exposure to lactate in brain tissues causes acidic tissue injury. On the other hand, lower CSF lactate levels may suggest the impaired function of lactate transport through MCT1, MCT2, MCT3, and/or MCT4 in CPEs in patients with dementia. The specific mechanism of energy supply to the brain in patients with neurodegenerative diseases has not yet been elucidated.

Thyroid hormones are considered to be involved not only in in neurogenesis and neurodifferentiation, but also in cognitive functions. It is interesting that hippocampal neurons are considered to be affected by thyroid hormone levels [75]. It has been pointed out that hypothyroidism is frequently associated with cognitive impairment and/or depressive-like behavior [76]. At present, however, specific foci responsible for symptoms of brain disorders, such as cognitive impairment, in patients with hypothyroidism, remain to be clarified. In addition, it is unclear why hippocampal neurons are affected in hypothyroidism. A large-scale study [77] showed that patients with hypothyroidism had a higher risk of memory impairment and also had a more than three-fold increase in the dementia risk. Several studies have reported an association between thyroid disorders and AD. However, there remains no consensus regarding the precise role of thyroid dysfunction in AD. A meta-analysis using clinical subject headings and keywords from databases [78] showed no significant association of hypothyroidism and the risk of cognitive dysfunction without adjustment for vascular comorbidities. On the other hand, another meta-analysis using some databases [79] showed that hypothyroidism was significantly more prevalent in patients with AD than in controls. It is likely that these discrepancies in findings on hypothyroidism in patients with dementia are due to the multiple causes of cognitive impairment.

MCT8, an active transporter that transports thyroid hormones [51,52], was reported to be expressed in endothelial cells but also in CPEs [55,58]. Also in this paper, MCT8 immunoreactivity was shown in CPEs as well as endothelial cells. These findings suggest that thyroid hormones can be transported from the blood into the hippocampus through these cells. There are two hypothesized mechanisms for thyroid hormones to move out of CP into CSF: one is the secretion of thyroid hormones bound to CP-derived transthyretin, and the other is the efflux of thyroid hormones via thyroid hormone transporters in CPEs such as MCT8 [80], as shown in Figure 3. Although putative directions of thyroid hormones through CPEs are also indicated by dashed arrows in Figure 3, these cannot be confirmed.

## 4. Conclusions and Future Direction

This paper reviewed the distribution of MCTs as transporters for lactate and thyroid hormones in the brain, especially in endothelial cells and CPEs. Lactate and thyroid hormones are important for the maintenance of several kinds of brain function. It is likely that their insufficiency or excess exposure to them due to CP damage causes brain dysfunction. Recent developments in brain imaging have increased the capacity to diagnose brain diseases. Functional ^1^H magnetic resonance spectroscopy (fMRS) is becoming a powerful diagnostic tool for brain diseases [81]. It was reported that lactate evaluated with fMRS has the potential to be a new diagnostic and prognostic marker for AD [81]. As the mitochondrial glycolysis pathway is considered to be disrupted in many brain disorders, the activation of anaerobic glycolysis with increased lactate production must occur in the presence of various brain disorders. Along with the development of brain imaging, the importance of lactate measurement will likely increase.

## Figures and Tables

**Figure 1 pharmaceutics-15-02062-f001:**
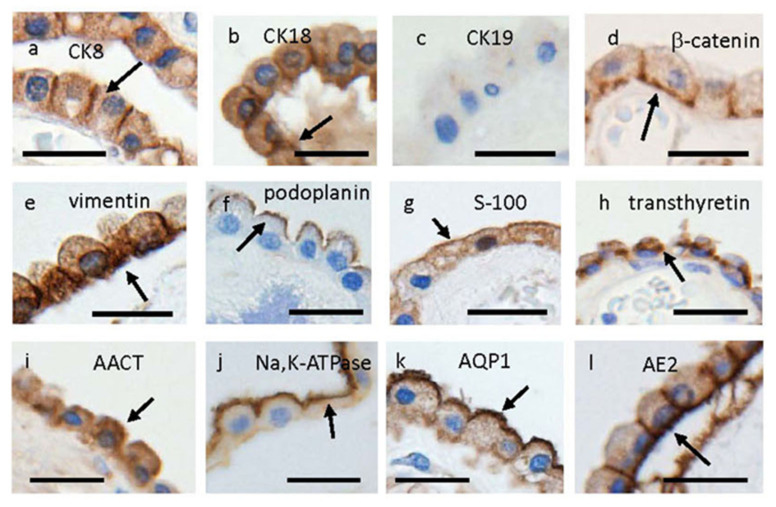
Distribution of immunoreactivities for CK8 (**a**), CK18 (**b**), CK19 (**c**), β-catenin (**d**), vimentin (**e**), podoplanin (**f**), S-100 (**g**), transthyretin/prealbumin (**h**), α1-antichymotrypsin (AACT) (**i**), Na^+^, -K^+^, -ATPase (**j**), AQP-1 (**k**), and AE2 (**l**), in epithelial cells of CP located in lateral ventricles of autopsied human brains. Arrows indicate immunoreactivity for these substances. Immunohistochemical findings from cases 2 (**i**), 3 (**d**,**g**), 6 (**a**,**k**), 7 (**f**,**h**), and 8 (**b,c**,**e**,**j**,**l**) are shown. Scale bars indicate 20 μm.

**Figure 2 pharmaceutics-15-02062-f002:**
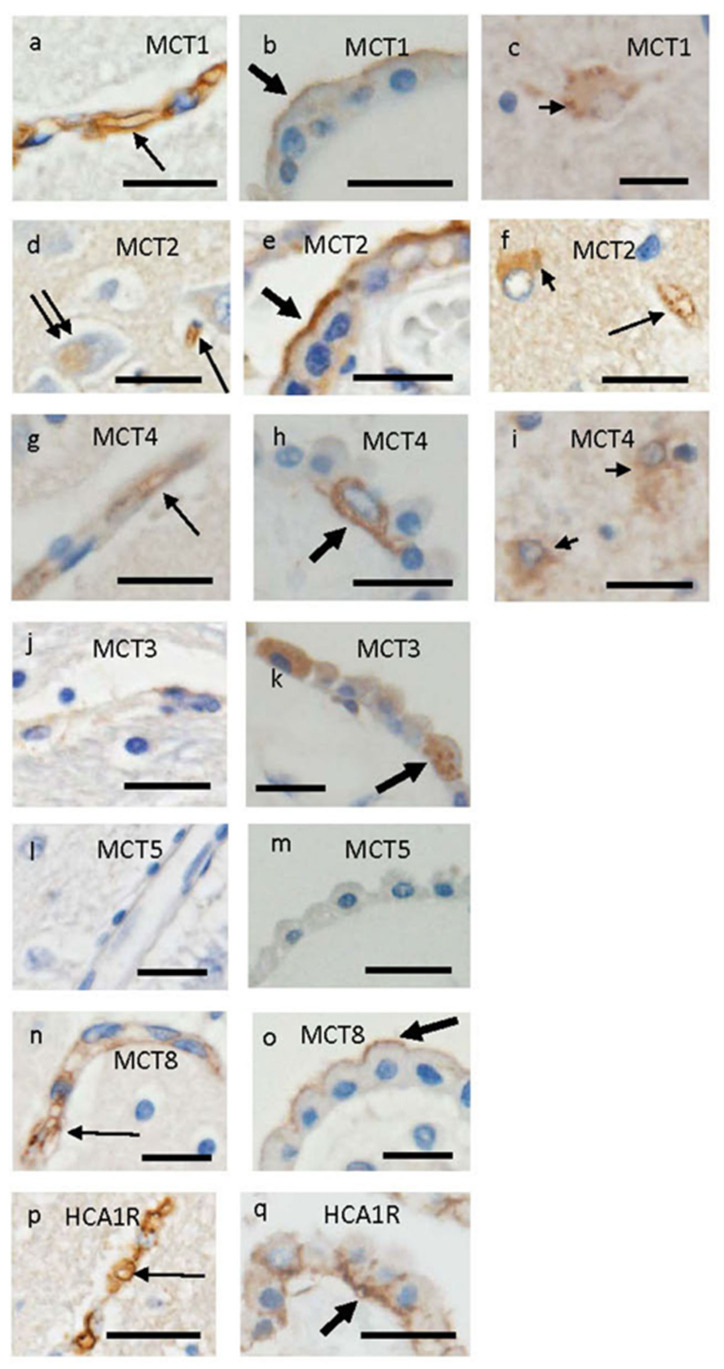
Distribution of immunoreactivities for MCT1 (**a**–**c**), MCT2 (**d**–**f**), MCT4 (**g**–**i**), MCT3 (**j**,**k**), MCT5 (**l**,**m**), MCT8 (**n**,**o**), and HCA1-R (**p**,**q**), in autopsied human brains. Immunohistochemical findings from cases 1 (**g**), 3 (**a**,**h**,**j**,**k**,**m**,**p**), 4, (**c**,**f**,**i**), 5 (**b**,**d**,**l**), 6 (**e**,**q**), and 8 (**n**,**o**) are shown. Immunohistochemical images in microvessels indicated by thin arrows in hippocampal samples are shown in (**a**,**d**,**f**,**g**,**n**,**p**), whereas images in epithelial cells of CP located in lateral ventricles indicated by thick arrows are shown in (**b**,**e**,**h**,**k**,**o**,**q**). Immunohistochemical images in reactive astrocytes in hippocampal samples indicated by short arrows are shown in (**c**,**f**,**i**), whereas MCT2 immunostaining in neurons in hippocampal samples indicated by double arrows is shown in (**d**). No MCT3 immunostaining in microvessels is shown in (**j**), whereas no MCT5 immunostaining in microvessels and CPEs is shown in (**l**,**m**). Scale bars indicate 20 μm.

**Figure 3 pharmaceutics-15-02062-f003:**
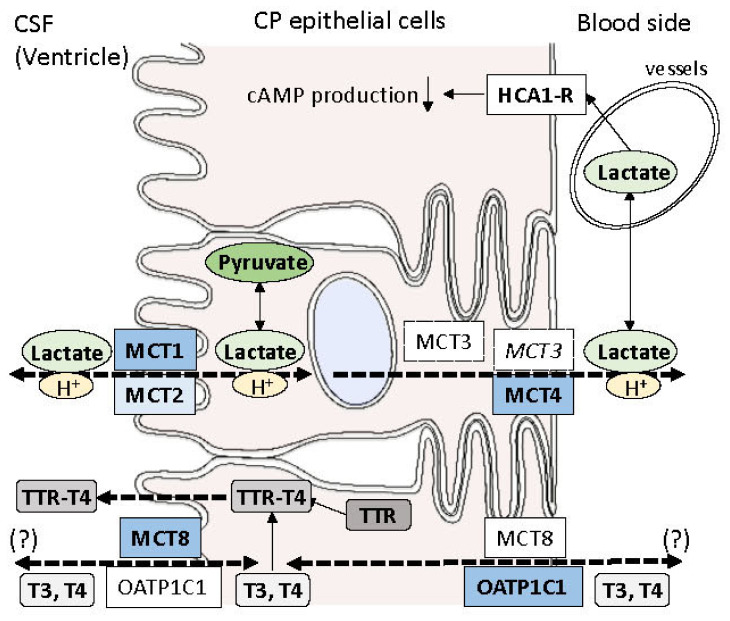
Polarized distribution of a receptor for lactate and transporters for lactate and thyroid hormones in CPEs. HCA1-R is expressed in the basolateral membrane of the cytoplasm of CPEs and induces decreased cyclic AMP production in the CPE cytoplasm. MCT1 and MCT2 are present on the apical (CSF-facing) side of CPEs, whereas MCT4 is present on the basal (CP stroma-facing) side of CPEs. MCT3 expression on the basolateral side of CPEs has been reported only in mice [34] but has not been confirmed in human brains, including CPEs [36]. Accordingly, MCT3 is written in italics and surrounded by dotted square lines. Putative directions of lactate through CPEs are indicated by dashed arrows. MCT8 and OATP1C1, which are known to be transporters for thyroid hormones, are considered to be distributed on the apical and basolateral sides of CPEs [55]. Thyroid hormones are considered to move between CSF and the CPE cytoplasm via these transporters. According to the paper reported by Roberts et al. [55], MCT8 is distributed on the apical surface of CPEs, whereas OATP1C1 is present primarily on the basolateral side of CPEs. Putative directions of thyroid hormones through CPEs are indicated by dashed arrows. However, the directions cannot be confirmed. As transthyretin (TTR) is synthesized in CPEs, T4 bound to TTR (TTR-T4) is considered to move from the cytoplasm of CPEs into ventricles.

**Table 1 pharmaceutics-15-02062-t001:** Summary of clinicopathological profiles.

(No.)	Age/Sex	Main Diagnosis
1	60/M	Dissecting aneurysm
2	64/M	Multiple system atrophy, Pneumonia
3	70/M	Myocardial infarction
4	71/M	Cerebellar tuberculosis
5	72/F	Pneumonia
6	74/M	Lung cancer
7	75/M	Gastric cancer
8	84/M	Myocardial infarction, Cerebral infarction

**Table 2 pharmaceutics-15-02062-t002:** Summary of antibodies used.

Antibody	Cat. No. (Clone Name)	Host Species and Usage
CK8	Progen, 61038	mouse, 1:50 (¶2)
CK18	ProteinTech, 66187-1-lg	mouse, 1:600 (¶2)
CK19	Progen, 61010	mouse, 1:10 (¶1)
β-catenin	SantaCruz, sc-7199	rabbit, 1:50 (-)
vimentin	DAKO, M0725 (V9)	mouse, 1:50 (¶1)
D2/40(podoplanin)	DAKO, M3619	mouse, 1:50 (¶1)
s-100	Nichirei, 422091	rabbit, diluted (-)
transthyretin	ProteinTech, 11891-1-AP	rabbit, 1:100 (-)
AACT	ProteinTech, 66078-1-lg	mouse, 1:500 (¶2)
Na^+^,-K^+^,-ATPase	SantaCruz, sc-48345	rabbit, 1:100 (¶1)
Aquaporin-1	ProteinTech, 20333-1-AP	rabbit, 1:250 (¶2)
AE2	sc-376632 (D-3)	mouse, 1:100 (¶1)
MCT1	Abcam, ab90582	mouse, 1:100 (¶1)
MCT2	Abcam, ab198272	rabbit, 1:100 (-)
MCT3	Abcam, ab60333	rabbit, 1:200 (-)
MCT4	Abcam, ab244385	rabbit, 1:50 (¶1)
MCT5	Abcam, ab191008	rabbit, 1:500 (¶1)
MCT8	Novus, NBP2-57308	rabbit, 1:200 (¶1)
HCA1-R	Novus, NLS-2095	rabbit, 1:200 (¶1)

(¶1, ¶2): Antigen retrieval with citrate buffer (pH 6) or Tris-EDTA buffer (pH 9) is needed prior to the application of the primary antibody. AACT: α1-antichymotrypsin.

**Table 3 pharmaceutics-15-02062-t003:** Regional distribution and cellular localization of MCTs in the brain.

Isoform (Gene)	Predominat Substrates	Regional Distribution	Cellular Localization	References
MCT1 (*SLC16A1*)	Lactate, pyruvate,	Widespread	Endothelial cells, astrocytes	[24,27,30,33,39,40,41,42,43,44,45]
	ketone bodies	Cortex, hippocampus,	ependymocytes, microglia,	
		cerebellum,	oligodensrocyte,	
		choroid plexus	choroid plexus epithelium,	
			*some neurons (Rt) ^#1^*	
MCT2 (*SLC16A7*)	Pyruvate, lactate,	Widespread	Neurons/axon, microglia,	[26,27,41,45]
	ketone bodies	Cortex, hippocampus,	endothelial cells,	
	(high affinity)	cerebellum,	choroid plexus epithelium	
		choroid plexus	*astrocytes (Rt, MS) ^#2^*	
MCT3 (*SLC16A8*)	Lactate	Localized	*Retinal pigment epithelium (Ms) ^#3^*,	[26,34,35]
			*choroid plexus epithelium (Ms) ^#3^*	
MCT4 (*SLC16A3*)	Lactate, pyruvate,	Widespread	Astrocytes, microglia	[26,27,41,43,46]
	ketone bodies	Cortex, hippocampus,	endothelial cells,	
	(low affinity)	cerebellum,	choroid plexus epithelium	
	(high capacity)	choroid plexus		
MCT5 (*SLC16A4*)	Orphan	Localized	Isolated choroid plexus	[26,36]
MCT8 (*SLC16A2*)	Thyroid hormone	Widespread	Neurons, astrocytes,	[26,53,54,55,56,57]
	(high affinity)	Cortex, hippocampus,	endothelial cells	
		hypothalamus,	choroid plexus epithelium	
		choroid plexus		

Italics indicate findings in rodents or humans with neurodegenerative diseases. #1: MCT1 expression is detected in some neurons of rats (Rt) [39]. #2: MCT2 expression is detected in end-feet of astrocytes in rats (Rt) [45] and astrocytes in brains in the presence of multiple sclerosis (MS) [41]. #3: MCT3 expression is detected in retinal pigment and choroid plexus epithelia of mice (Ms) [34].

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
