# Peer review of "Distribution of Monocarboxylate Transporters in Brain and Choroid Plexus Epithelium"

_pharmaceutics, 2023, doi:10.3390/pharmaceutics15082062_

Round 1

Reviewer 1 Report

This interesting review is devoted to describing the localization of monocarboxylate transporters in brain and choroid plexus epithelium.

Comments:

1.      There are figures in the article, but it is not very clear from the current description whether the figures come from previously published works or are original. If they are figures from published works, it is recommended to add references. If they are original figures, their description is not sufficient. It is recommended to add more data about how the data were obtained (description of patients, whether ethical approval was obtained, etc.).

2.      It is recommended to strengthen the description of the clinical significance and perspectives of the data described in the review.

Author Response

<To comments of reviewer 1>

  • There are figures in the article, but it is not very clear from the current description whether the figures come from previously published works or are original. If they are figures from published works, it is recommended to add references. If they are original figures, their description is not sufficient. It is recommended to add more data about how the data were obtained (description of patients, whether ethical approval was obtained, etc.).

(to the comment) Previously published findings were summarized in Table 3. Immunohistochemical images in Figs.1 &2 were used to confirm the findings. Accordingly, Tables 1 & 2 were added to explain clinicopathological information on autopsied brain samples and antibodies used to confirm them. In addition, descriptions on ethical approval were added on lines 87-91. Two references [24, 25] were added to prove ethical approval.

  • It is recommended to strengthen the description of the clinical significance and perspectives of the data described in the review.

(to the comment) Descriptions on the clinical significance and perspectives of the data were added on lines 127-130, 291-294, and 350-358.

Reviewer 2 Report

The review is timely and well written. However the authors should pay attention to the figures and add tables to improve the quality of the presentation of their work.

1) The references for BBB are not well chosen, the authors should use references from authors specialized on BBB, not their own work.

2) Figure 1 and 2. The legends are difficult to read without reference to the  text. The reviewer suggest to compose a table with all the relevant info and corresponding literature references so that for each panel of the figure one could refer to the table info. The corresponding text will so simplify helping the reader for understanding.

3) Figure 3 is of really low quality, it should be recomposed. Introducing colors will greatly help.

4) The conclusion should be developed

Author Response

<To comments of reviewer 2>

The review is timely and well written. However the authors should pay attention to the figures and add tables to improve the quality of the presentation of their work.

(to the comment) We changed three figures and Table 1 to new Table 3 and added new Table 1 and Table 2 to improve the quality of the presentation.

  • The references for BBB are not well chosen, the authors should use references from authors specialized on BBB, not their own work.

(to the comment) According to the reiewer’s comment, we added 3 references, which were well-known representative or recently published, on BBB [5-7].

  • Figure 1 and 2. The legends are difficult to read without reference to the text. The reviewer suggest to compose a table with all the relevant info and corresponding literature references so that for each panel of the figure one could refer to the table info. The corresponding text will so simplify helping the reader for understanding.

(to the comment) Tables 1 & 2 were added to explain clinicopathological information on autopsied brain samples and antibodies antibodies used to confirm previously published findings.

  • Figure 3 is of really low quality, it should be recomposed. Introducing colors will greatly help.

(to the comment) We changed Figure 3 to new one. Various colors are used in new Fig. 3.

  • The conclusion should be developed.

(to the comment) We added 5 sentences in Conclusion and Future Direction to develop the conclusion (Lines 350-358).

Reviewer 3 Report

This manuscript by Ueno et al. is a literature review on the distribution of MCTs in the CNS. The authors have also confirmed the localization of some choroid plexus epithelial cell markers and MCTs using immunostainings.

I have the following comments/questions:

Line 24: I would suggest changing the word “scanty” for something more scientific.

Line 34: “The” is missing at the beginning of the sentence.

Line 35: there are two basement membranes, membranes should be plural

Line 36: I would suggest writing: “Blood-brain barrier endothelial cells (BBB-ECs) have few vesicles….” Otherwise, the current sentence means that a few specific vesicles are characterizing BBB-ECs.

Line 37: The nomenclature used to describe parts of the choroid plexus (stroma and parenchyma) seem to be use interchangeably in the literature. I would suggest to the author to add a cartoon labeling the different parts described.

Line 39: stroma of “the” CP.

Line 39: I would suggest rewriting “CP with a vascularized stroma…” Every choroid plexus has a highly vascularized stroma.

Line 41: the abbreviation (CPEs) should be mentioned here first and not in the following paragraph.

Line 47: I suggest changing “localization” for “presence” or “expression”

Line 51: The abbreviation CK should be introduced here

Line 53: I would suggest making it clearer that the authors themselves are confirming the data in this manuscript. For example: “We couldn’t confirm the immunoreactivity for CK19 …”. In addition, it should be mentioned that the stainings were done in human CPs.

Figure 1 and 2: The figure legend should mention the origin of the tissue (human brain), which choroid plexus were used, ie. from which ventricle. Are there known differences (for example in protein expression) between choroid plexus found in the different ventricles within an individual?  Additionally, the authors should mention if the tissue was otherwise normal; what was the cause of death, age of the patient? The figure legends should also mention what the arrows are pointing to and if the images show a section of a choroid plexus or some other tissue. Finally, scale bars should be added to the images.

Line 58: “immunoreactivities” isn’t often used in scientific papers. I would suggest: “We confirm, as shown here by immunostainings, that B-catenin….”

Line 64: The authors should make sure that the manuscript font and spacing is the same throughout.

Line 67: functions of “the” CP

Line 77: expression at “the” BCSFB in the “context” of neurological diseases.

Line 87: “the” is missing in the title before brain

Line 101: replace “in the” by “of”

Line 104-108: I would suggest re-writing this section to improve clarity.

Line 110: the authors are submitting a review manuscript; it is not a study.

Line 126: “the” is missing before brain in the title of this section

Line 131: it is not clear what the authors mean by “oligo may be taken up as extracellular lactate”

Line 130-131, Line 137-138 and Line 138-139: These three sentences all mention the same thing without bringing additional information. I would suggest to the authors to combine them or explicitly state the differences in the study that warrants mentioning them all, ie. difference in tissue, staining techniques, etc.

Line 133: there is no link with the previous sentence

Line 151-155: similar to the previous paragraph, I would suggest to the authors to make it clearer which cells express MCT2 and which one expresses it the most. The same applies to section 2.3.

Line 156: Immunoreactivity for MCT2 and using an antibody against MCT2 is somewhat redundant.

Line 177: again it would be better if the authors mentioned which type of tissue they are using.

Line 180: “was introduced” should be changed to “was first reported”

Discussion: Again, I would suggest to the authors to make it clear what as been shown by themselves in this manuscript versus what was shown previously by others.  

Final thoughts: The manuscript by Ueno et al. is of interest, however, the manuscript would require moderate editing to improve the grammar and syntax of the manuscript. I would suggest using the help of a native English speaker to correct phrasing and improve the overall manuscript.

However, the main issue is that plagiarism was detected. For example, line 227 to line 234 is a verbatim copy of a manuscript previously published by the authors (https://doi.org/10.1016/j.neulet.2020.135479). Even is the text was originally written by the authors, it should be rephrased in this manuscript. In addition, the authors tend to copy one or two sentences from one paper and the next sentence or two from another, changing only a few words from the original text. This isn’t acceptable. The offending text must be modified otherwise I will have to recommend the rejection of this manuscript.

see comments

Author Response

<To comments of reviewer 3>

This manuscript by Ueno et al. is a literature review on the distribution of MCTs in the CNS. The authors have also confirmed the localization of some choroid plexus epithelial cell markers and MCTs using immunostainings.

I have the following comments/questions:

Line 24: I would suggest changing the word “scanty” for something more scientific.

(to the comment) We changed “scanty” to “poor”.

Line 34: “The” is missing at the beginning of the sentence.

(to the comment) We added “The”.

Line 35: there are two basement membranes, membranes should be plural.

(to the comment) We changed to “two basement membranes”.

Line 36: I would suggest writing: “Blood-brain barrier endothelial cells (BBB-ECs) have few vesicles….” Otherwise, the current sentence means that a few specific vesicles are characterizing BBB-ECs.

(to the comment) We changed to “BBB endothelial cells have few vesicles”.

Line 37: The nomenclature used to describe parts of the choroid plexus (stroma and parenchyma) seem to be use interchangeably in the literature. I would suggest to the author to add a cartoon labeling the different parts described.

(to the comment) We unified to use “stroma” in this manuscript.

Line 39: stroma of “the” CP.

(to the comment) We added “the”.

Line 39: I would suggest rewriting “CP with a vascularized stroma…” Every choroid plexus has a highly vascularized stroma.

(to the comment) We rewrote “CP with a vascularized stroma”.

Line 41: the abbreviation (CPEs) should be mentioned here first and not in the following paragraph.

(to the comment) The abbreviation (CPEs) was mentioned on line 48 in the paragraph of revised manuscript (on line 41 in the original manuscript).

Line 47: I suggest changing “localization” for “presence” or “expression”

(to the comment) A word “localization” was changed to “presence”. The word “localization” in other sentences was also changed to “presence” or “expression”.

Line 51: The abbreviation CK should be introduced here

(to the comment) The abbreviation CK was introduced in the place (line 58 in the revised manuscript).

Line 53: I would suggest making it clearer that the authors themselves are confirming the data in this manuscript. For example: “We couldn’t confirm the immunoreactivity for CK19 …”. In addition, it should be mentioned that the stainings were done in human CPs.

(to the comment) We changed to “We couldn’t confirm the immunoreactivity”.

Figure 1 and 2: The figure legend should mention the origin of the tissue (human brain), which choroid plexus were used, ie. from which ventricle. Are there known differences (for example in protein expression) between choroid plexus found in the different ventricles within an individual?  Additionally, the authors should mention if the tissue was otherwise normal; what was the cause of death, age of the patient? The figure legends should also mention what the arrows are pointing to and if the images show a section of a choroid plexus or some other tissue. Finally, scale bars should be added to the images.

(to the comment) We changed Figs. 1 &2 and added new two tables as Tables 1 &2. In this manuscript, choroid plexus samples in lateral ventricles and hippocampal samples of autopsied human brains were used. The autopsied human brains were not normal. Various invasions including hypoxic and ischemic damage joined the brain by the time of death. There were no clear differences in immunohistochemical findings among ventricles. Clinicopathological findings were shown in Table1. Information on antibodies was shown in Table 2. Scale bars were added in Figs. 1 & 2. We added descriptions on cells indicated by various arrows.

Line 58: “immunoreactivities” isn’t often used in scientific papers. I would suggest: “We confirm, as shown here by immunostainings, that B-catenin….”

(to the comment) We changed to “We confirmed, as shown here by immunostainings, that”.

Line 64: The authors should make sure that the manuscript font and spacing is the same throughout.

(to the comment) We made sure that the manuscript font and spacing is the same throughout.

Line 67: functions of “the” CP

(to the comment) We added “the”.

Line 77: expression at “the” BCSFB in the “context” of neurological diseases.

(to the comment) We added “the”.

Line 87: “the” is missing in the title before brain

(to the comment) We added “the” in the title.

Line 101: replace “in the” by “of”

(to the comment) We replace “in the” by “of”.

Line 104-108: I would suggest re-writing this section to improve clarity.

(to the comment) We changed two sentences (lines 127-130).

Line 110: the authors are submitting a review manuscript; it is not a study.

(to the comment) We changed two sentences (lines 131-137).

Line 126: “the” is missing before brain in the title of this section

(to the comment) We added “the” in the title.

Line 131: it is not clear what the authors mean by “oligo may be taken up as extracellular lactate”

(to the comment) We changed this sentence such as “may taken up into oligodendrocytes” (lines 159-161).

Line 130-131, Line 137-138 and Line 138-139: These three sentences all mention the same thing without bringing additional information. I would suggest to the authors to combine them or explicitly state the differences in the study that warrants mentioning them all, ie. difference in tissue, staining techniques, etc.

(to the comment) According to the reviewer’s comments, we summarized what are written in three sentences and changed four sentences (lines 157-165).

Line 133: there is no link with the previous sentence

(to the comment) One sentence on prion proteins was deleted. Accordingly, one reference on the prion proteins [38] in the original manuscript was deleted

Line 151-155: similar to the previous paragraph, I would suggest to the authors to make it clearer which cells express MCT2 and which one expresses it the most. The same applies to section 2.3.

(to the comment) According to the reviewer’s comment, 4 sentences were changed (lines 176-184).

Line 156: Immunoreactivity for MCT2 and using an antibody against MCT2 is somewhat redundant.

(to the comment) One phrase “using an antibody for MCT2” was deleted.

Line 177: again it would be better if the authors mentioned which type of tissue they are using.

(to the comment) We added one word “human”.

Line 180: “was introduced” should be changed to “was first reported”

(to the comment) One phrase “was introduced” was changed to “was first reported”.

Discussion: Again, I would suggest to the authors to make it clear what as been shown by themselves in this manuscript versus what was shown previously by others. 

(to the comment) We made photos using autopsied human brains and images of immunostainings were used in Figs. 1 & 2. Previously reported findings in papers were summarized in Table 3. One sentence (lines 243-245) was added to explain it.

Final thoughts: The manuscript by Ueno et al. is of interest, however, the manuscript would require moderate editing to improve the grammar and syntax of the manuscript. I would suggest using the help of a native English speaker to correct phrasing and improve the overall manuscript.

(to the comment) This revised manuscript was widely corrected.

However, the main issue is that plagiarism was detected. For example, line 227 to line 234 is a verbatim copy of a manuscript previously published by the authors (https://doi.org/10.1016/j.neulet.2020.135479). Even is the text was originally written by the authors, it should be rephrased in this manuscript. In addition, the authors tend to copy one or two sentences from one paper and the next sentence or two from another, changing only a few words from the original text. This isn’t acceptable. The offending text must be modified otherwise I will have to recommend the rejection of this manuscript.

(to the comment) Thank you so much. Your points are very important. Three sentences on lines 227-234 in the original manuscript were totally changed to five sentences in the revised manuscript (lines 254-260). In addition, according to the reviewer’s comments, some phrases and sentences were changed (lines 41-43, 82-83, 176-180, 232-234, 270-280, 315-316, 332-333).

Reviewer 4 Report

The focus of this review is the distribution and localization of monocarboxylate transporters in brain and choroid plexus epithelia. It is well organized and well written and is helpful for the understanding how lactate is distributed and used in the brain as an energy source. I have the following minor suggestions:

- Figure 1 and 2 show immunostaining of marker proteins and MCTs. These results don't seem to be published previously. If this is the case, the methods for the immunostaining need to be described in more detail.

- Figure 2 shows immunostaining of MCTs in different brain tissues/cells. The legend doesn't contain the indication which panels correpond to which cell types.

- For figure 1 and 2, it also needs to be pointed out whether these are results obtained with human and rat tissue samples

-  For the understanding of the flow of lactate in brain, it is necessary to know not only the distribution and localization of the transporters, but also the kinetic parameters. As the authors pointed out in the review that some transporters are high affinity and some low affinity transporters for lactate. A table indicating the affinity of the individual transporters for lactate would be helpful and make the review more comprehensive.

Moderate English editing is necessary. Here a few examples:

- Line 104, "It is meaningful to advance understanding ...". Not sure what this sentence means

- Line 131, "Rinholm et al. [37] reported that oligodendrocytes may be taken up ...". It should be "oligodendrocytes may take up ..."

Author Response

<To reviewer 4>

The focus of this review is the distribution and localization of monocarboxylate transporters in brain and choroid plexus epithelia. It is well organized and well written and is helpful for the understanding how lactate is distributed and used in the brain as an energy source. I have the following minor suggestions:

- Figure 1 and 2 show immunostaining of marker proteins and MCTs. These results don't seem to be published previously. If this is the case, the methods for the immunostaining need to be described in more detail.

(to the comment) Tables 1 &2 were newly added to explain human samples used in this manuscript and immunohistochemical stainings performed to confirm previously reported findings.

- Figure 2 shows immunostaining of MCTs in different brain tissues/cells. The legend doesn't contain the indication which panels correpond to which cell types.

(to the comment) A new legend of Fig. 2 contained the indication which panels correspond to which cell types. Immunostainings of vessels are shown in (a, d, g, j, l, n, and p), whereas those of CPEs are shown in (b, e, h, k, m, o, and q). Immunostaining of reactive astrocytes are shown in (c, f, and i), whereas immunostaining of neurons is shown in (d). Vessels in (j, l) and epithelial cells in (m) were not stained.

- For figure 1 and 2, it also needs to be pointed out whether these are results obtained with human and rat tissue samples

(to the comment) As only human brain samples were used in Figs. 1 & 2 in this review manuscript, descriptions on species were added in legends of Figs. 1 & 2.

-  For the understanding of the flow of lactate in brain, it is necessary to know not only the distribution and localization of the transporters, but also the kinetic parameters. As the authors pointed out in the review that some transporters are high affinity and some low affinity transporters for lactate. A table indicating the affinity of the individual transporters for lactate would be helpful and make the review more comprehensive.

(to the comment) Some words on affinity and capacity were added in Table 3.

Comments on the Quality of English Language

Moderate English editing is necessary. Here a few examples:- Line 104, "It is meaningful to advance understanding ...". Not sure what this sentence means- Line 131, "Rinholm et al. [37] reported that oligodendrocytes may be taken up ...". It should be "oligodendrocytes may take up ..."

(to the comments) Two sentences in the original manuscript (lines 104-108, lines 131-134) were totally changed to two sentences in the revised manuscript (lines 127-130, lines 159-161), respectively

Reviewer 5 Report

In this study, Ueno et al., reviewed the recent finding regarding the distribution and significance of MCTs in the brain, especially in CPEs. Although interesting, additional information is encouraged to supply.

1.     The authors didn't mention what are these samples in Figure 1 and Figure 2, are they from rodent or human? If these figures are come from other research, please indicate clearly. 

2.     Also, are these samples from normal brain or diseased brain? If in brain with neurodegenerative disease, such as AD or PD, what is the expression level of MCT in CPEs?

3.     The authors should focus on the specific function of different MCTs expressed on CPEs.

4.     Why the authors only focus on lactate transport particularly? Other substates of MCTs should also be reviewed and summarized.

5.     The correlation between expression of MCTs in CPEs with brain diseases is weakly discussed, I would suggest providing more detail evidence and discussion.

Author Response

<To reviewer 5>

In this study, Ueno et al., reviewed the recent finding regarding the distribution and significance of MCTs in the brain, especially in CPEs. Although interesting, additional information is encouraged to supply.

  • The authors didn't mention what are these samples in Figure 1 and Figure 2, are they from rodent or human? If these figures are come from other research, please indicate clearly. 

(to the comment) We used only human samples to confirm previously reported results. Accordingly, Table 1 was added. Immunohistochemical images used in Figs. 1 & 2 were not published in other papers, although staining results have been already reported.

  • Also, are these samples from normal brain or diseased brain? If in brain with neurodegenerative disease, such as AD or PD, what is the expression level of MCT in CPEs?

(to the comment) Autopsied human brains used in this review manuscript were not normal and various invasions including hypoxic and ischemic damage joined the brain by the time of death. Although brains with AD and PD were not included, one brain (No.2) with multiple system atrophy was included in autopsied human brains used in this manuscript. There were no clear differences in localization of immunostaining among 8 human brains. On the other hand, it is very interesting whether the expression level of MCT in CPEs is increased or decreased in AD and PD. Unfortunately, it seems that it has not yet been elucidated.

  • The authors should focus on the specific function of different MCTs expressed on CPEs.

(to the comment) We added descriptions on function of MCTs in Discussion (lines 291-294) and Figure 3.

  • Why the authors only focus on lactate transport particularly? Other substates of MCTs should also be reviewed and summarized.

In this manuscript, as we tried to investigate the distribution of MCTs which were previously reported to be present in CPEs, MCT1, MCT2, MCT3, MCT4, and MCT5 were first reviewed. MCTs are important transporters for lactate in CPEs as well as neurons, astrocytes, and microvessels. In addition, as MCT8, a transporter for thyroid hormone, was also known to be expressed in CPEs, the distribution of MCT8 was reviewed. MCTs reported previously to be expressed in CPEs are reviewed in this manuscript.

  • The correlation between expression of MCTs in CPEs with brain diseases is weakly discussed, I would suggest providing more detail evidence and discussion.

At present, information on expression of MCTs in CPEs with brain diseases is poor. We added five sentences on clinical significance of lactate concentration in the brain (lines 349-358). As the mitochondrial glycolysis pathway is supposed to be disrupted in many brain disorders, the activation of anaerobic glycolysis with increased lactate production must be happening in various brain disorders. New insights on MCTs in brains with neurodegenerative diseases are expected.

Round 2

Reviewer 2 Report

The authors revised their manuscript in a satisfactory manner and adopted all the suggested possible improvements. The manuscript is now acceptable

Author Response

Thank you for your comments. I additionally revised our manuscript, according to another reviewer.

Reviewer 3 Report

The authors have responded or addressed most of my comments. However, the quality of the English language used is still subpar. While readers would be able to understand the manuscript, getting help from a proofreader would dramatically improve the manuscript. 

Finally, a few sections flagged as plagiarism must still be modified:

Line 77-83, Line 159-166, Line 299-305, Line 681-683, Line 688-692.  

see above

Author Response

A native English speaker has gone over this manuscript again.

Many sections including the sections mentioned by the reviewer were changed to another words, phrases, or sentences.

Round 3

Reviewer 3 Report

The authors have not made a single change in the sections I specifically identified. See previous comment bellow.

Finally, a few sections flagged as plagiarism must still be modified:

Line 77-83, Line 159-166, Line 299-305, Line 681-683, Line 688-692

Keep in mind that these are the line numbers from version 2 of the manuscript. 

Author Response

<Comments of reviewer 3>

The authors have not made a single change in the sections I specifically identified. See previous comment bellow.

Finally, a few sections flagged as plagiarism must still be modified:

Line 77-83, Line 159-166, Line 299-305, Line 681-683, Line 688-692.

Keep in mind that these are the line numbers from version 2 of the manuscript. 

(to the comments)

In this revised version, many words, phrases, and sentences were revised by green letters. Five sections (Line 77-83, Line 159-166, Line 299-305, Line 681-683, Line 688-692) mentioned by the reviewer 3 were changed to another words, phrases, or sentences (Line 74-81, Line 126-134, Line 174-179, Line 324-326, Line 331-337) in this revised version, respectively. Many points shown by green letters other than the above points were revised.

Round 4

Reviewer 3 Report

The authors have made the necessary modifications.

Minor errors were detected. The manuscript has been improved compared to the initial submission.